# Organisational Culture and Mask-Wearing Practices for Tuberculosis Infection Prevention and Control among Health Care Workers in Primary Care Facilities in the Western Cape, South Africa: A Qualitative Study

**DOI:** 10.3390/ijerph182212133

**Published:** 2021-11-19

**Authors:** Idriss I. Kallon, Alison Swartz, Christopher J. Colvin, Hayley MacGregor, Gimenne Zwama, Anna S. Voce, Alison D. Grant, Karina Kielmann

**Affiliations:** 1Division of Social and Behavioural Sciences, School of Public Health and Family Medicine, Faculty of Health Sciences, University of Cape Town, Cape Town 7925, South Africa; kalloni@sun.ac.za (I.I.K.); alison.swartz@uct.ac.za (A.S.); 2Centre for Evidence-Based Health Care, Division of Epidemiology and Biostatistics, Department of Global Health, Faculty of Medicine and Health Sciences, Stellenbosch University, Cape Town 7505, South Africa; 3Department of Epidemiology, School of Public Health, Brown University, Providence, RI 02912, USA; 4Department of Public Health Sciences, School of Medicine, University of Virginia, Charlottesville, VA 22903, USA; 5Institute of Development Studies, University of Sussex, Brighton BN1 9RE, UK; h.macgregor@ids.ac.uk; 6Institute of Global Health & Development, Queen Margaret University, Musselburgh EH21 6UU, UK; GZwama@qmu.ac.uk (G.Z.); kkielmann@qmu.ac.uk (K.K.); 7School of Nursing and Public Health, College of Health Sciences, University of KwaZulu-Natal, Durban 4041, South Africa; Voceas@ukzn.ac.za; 8TB Centre, London School of Hygiene & Tropical Medicine, London WC1E 7HT, UK; alison.grant@lshtm.ac.uk; 9Africa Health Research Institute, School of Laboratory Medicine and Medical Sciences, College of Health Sciences, University of KwaZulu-Natal, Durban 4001, South Africa; 10Department of Public Health, Institute of Tropical Medicine, 2000 Antwerp, Belgium

**Keywords:** infection prevention and control, tuberculosis, PPE, masks, organisational culture, South Africa

## Abstract

*Background*: Although many healthcare workers (HCWs) are aware of the protective role that mask-wearing has in reducing transmission of tuberculosis (TB) and other airborne diseases, studies on infection prevention and control (IPC) for TB in South Africa indicate that mask-wearing is often poorly implemented. Mask-wearing practices are influenced by aspects of the environment and organisational culture within which HCWs work. *Methods*: We draw on 23 interviews and four focus group discussions conducted with 44 HCWs in six primary care facilities in the Western Cape Province of South Africa. Three key dimensions of organisational culture were used to guide a thematic analysis of HCWs’ perceptions of masks and mask-wearing practices in the context of TB infection prevention and control. *Results*: First, HCW accounts address both the physical experience of wearing masks, as well as how mask-wearing is perceived in social interactions, reflecting visual manifestations of organisational culture in clinics. Second, HCWs expressed shared ways of thinking in their normalisation of TB as an inevitable risk that is inherent to their work and their localization of TB risk in specific areas of the clinic. Third, deeper assumptions about mask-wearing as an individual choice rather than a collective responsibility were embedded in power and accountability relationships among HCWs and clinic managers. These features of organisational culture are underpinned by broader systemic shortcomings, including limited availability of masks, poorly enforced protocols, and a general lack of role modelling around mask-wearing. HCW mask-wearing was thus shaped not only by individual knowledge and motivation but also by the embodied social dimensions of mask-wearing, the perceptions that TB risk was normal and localizable, and a shared underlying tendency to assume that mask-wearing, ultimately, was a matter of individual choice and responsibility. *Conclusions*: Organisational culture has an important, and under-researched, impact on HCW mask-wearing and other PPE and IPC practices. Consistent mask-wearing might become a more routine feature of IPC in health facilities if facility managers more actively promote engagement with TB-IPC guidelines and develop a sense of collective involvement and ownership of TB-IPC in facilities.

## 1. Introduction

This paper examines mask-wearing among healthcare workers (HCWs) in South African primary care clinics in a context with high HIV prevalence and high occupational exposure and risk related to TB. While the fieldwork for our study was conducted before the COVID-19 pandemic, the global spread of COVID-19 has highlighted the critical importance of preventing the transmission of infectious diseases and ensuring prevention practices are guided by accurate perceptions of risk. Mask-wearing (unless specified otherwise, “mask” here refers to both surgical masks and N95 respirators) is often the focus of personal risk reduction strategies and has received a great deal of media and academic attention during the COVID-19 pandemic. Indeed, mask-wearing, along with the use of other personal protective equipment (PPE), is often the most visible aspect of infection prevention and control (IPC) strategies in clinics and hospitals. However, within health care settings, mask-wearing is not a new practice but rather one of several key control measures that has been long recommended for prevention of transmission of infectious diseases, such as tuberculosis (TB), to health care staff and patients in health facilities [1]. N95 respirators are a highly effective IPC measure for respiratory infections [2]. Although less protective than N95 respirators, medical or surgical masks have also been found to be useful in reducing the risk of clinical respiratory infections and influenza-like illness [3]. Medical or surgical masks—the ones mostly used by patients in South African health facilities—can help in protecting other people from large respiratory droplets. However, they do not protect users from airborne infection because they vary in “thickness and permeability” compared to N95 respirators [3].

The potential for TB transmission in both hospital and primary care is of significant concern in South Africa [4,5] given the extremely high levels of TB in the country as well as the sub-optimal screening and diagnosis of people with TB-related symptoms [6,7]. In one South African district, for example, primary health care (PHC) clinics failed to optimally screen between 63% and 79% of people with TB-related symptoms, and between 90% and 100% of those attending clinics for other reasons [5]. Recent evidence also shows that HCWs are at heightened risk of TB [1,8]. In some South African settings, they are two to three times more likely to have TB than the general population [6,8]. However, while many HCWs are aware of the important protective role that mask-wearing has in reducing transmission of TB and other airborne illnesses, several studies on TB-IPC in South Africa indicate that mask-wearing is often poorly implemented [9,10,11,12].

Most studies documenting ‘poor compliance’ with N95 respirators tend to focus solely on issues with individual HCW decision-making, knowledge, or motivation [13]. However, research has also identified other important barriers, including the physical discomfort of the protective wear itself [14], as well as the social and symbolic meanings of masks [14,15]. These studies demonstrate that mask-wearing, particularly N95 respirators for HCWs, is profoundly affected by the social and embodied experience of mask-wearing and the working environment within which HCWs work. These factors include the materiality of the mask itself, the type of work a cadre of health care staff does, and shared ways of thinking about infection prevention and control. Clinic infrastructure also plays an important role in TB-IPC. In South Africa, some clinics were upgraded, and new clinics were built after the end of apartheid, but others have not been refurbished for decades [16]. The built environment within clinics can have a significant effect on both TB transmission risk as well as the ease and comfort of implementing TB-IPC protocols [17,18]. Our understanding of how work environments, clinic infrastructures, and professional practices influence TB-IPC implementation is, however, quite limited [19].

The idea that organizations have their own cultural context that shapes working norms and practices offers an important lens through which to understand how the perceptions and management of disease transmission risk operate within the day-to-day operation of the clinic [15,20]. To date, most studies of organisational culture in the IPC and health services literature have focused on hospital settings in high-income countries [21,22,23]. In this article, we document HCWs’ individual and collective risk perceptions and practices of mask-wearing within the broader organisational culture of selected PHC facilities in the Western Cape (WC), South Africa.

Definitions of organisational culture in healthcare often borrow familiar elements from broader models of culture from other disciplines, such as values, behaviours, dress, language, symbols, rituals, myths, and forms of authority [21,24]. Mannion and Davies [25] (p. 2) offer a useful framing of the elements of organisational culture in health care settings, describing them as the “softer, less visible, aspects of health service organisations and how these become manifest in patterns of care, as well the narratives that are used to explain what is done and why”. They describe three levels of organisational culture: (1) “visible manifestations” that include staff training activities, reporting and clinical practice protocols, structures of staff compensation, and procedures for managing clients’ safety and risk; (2) “shared ways of thinking” that include the ways staff talk with each other about the roles that they must perform, their perceptions of risk, and the meanings and experiences of their work; (3) the mostly hidden “deeper shared assumptions” that include HCWs’ beliefs about their own and their patients’ roles and responsibilities, as well as reflections on the social and power relations among and between them.

This paper uses the Mannion and Davies framework of organisational culture to inform our analysis of some of the underlying dynamics within clinics (pre-COVID) that shape mask-wearing practices in primary health facilities in the Western Cape Province of South Africa. The framework serves as a general guide for identifying shared practices, norms, and values around mask-wearing. This allows us to not only document the ways that organisational culture might work as a barrier to effective TB-IPC but also to consider how changes in organisational culture in clinics might support improved IPC in South Africa and more generally.

## 2. Methods

We draw on a sub-set of qualitative data collected as part of the interdisciplinary, mixed methods project *Umoya omuhle,* which can be roughly translated as “good air” in isiZulu. The project employed a whole-systems approach that addressed social, biological, and infrastructural factors to holistically study nosocomial transmission of DR-TB and TB-IPC implementation in two South African provinces, the WC and KwaZulu-Natal [26]. As part of the *Umoya omuhle* project, we examined health system influences on the implementation of WHO and the South African National Department of Health-recommended IPC practices, including the use of PPE. When using the term ‘mask’, our HCW participants most frequently referred to N95 respirators that they were expected to wear in health settings. Another important part of the context is that patients have been increasingly asked to wear surgical masks in primary care clinics and newly diagnosed TB patients may be asked to wear their own N95 respirators.

### 2.1. Study Sites: Sampling and Selection

Guided by the larger project objectives, we purposively sampled PHC facilities in the Western Cape (WC), coded in this article as WC1–WC6. The WC province has one of the highest TB burdens in the country, with an incidence of 591 per 100,000 population in 2019 [27], as well as significant variation between clinics with respect to reported mask-wearing, making it an appropriate location to study mask-wearing in closer context. Facilities were selected to ensure maximum variation in relation to facility location (urban/rural), physical infrastructure (i.e., built before or after a clinic infrastructure revitalization effort in the late 1980s), patient load and the organisation of care, including health services rendered and the presence of patient appointment systems for TB or other chronic conditions, or integrated care, particularly for HIV and TB (see Table 1 below).

Two of the selected facilities (WC2 and WC3) were located in peri-urban townships characterised by persistent challenges of poverty, high burden of disease, and lack of adequate access to basic services, including, housing, water, and sanitation. The other facilities (WC1 and WC4–6) were located in lower middle-income suburbs, with generally better access to housing and services. All health facilities were headed by a clinic or operational manager and supported by HCWs with defined clinical roles. All facilities offered several PHC services and made use of patient appointment systems in an attempt to reduce waiting and crowding in facilities. Smaller facilities, known as clinics, typically focus on care for children, and non-curative treatments, while community health centres (CHC) offer additional services, including physiotherapy, occupational therapy, and dieticians. Some CHCs, such as WC5, provide 24 h maternity, accident, and emergency services. Community day clinics (CDC), like WC4, also provide additional services, such as dentistry and psychiatry, but do not provide 24 h emergency services.

**Table 1 ijerph-18-12133-t001:** Facility descriptions and interviewees.

Facility	Location	No of HCW Staff at Facility	General Monthly Patient Head Count
WC1	Urban clinic built after 2010	43	2000–3000
WC2	Peri-urban clinic built after 2000	6	800–1500
WC3	Peri-urban clinic built in 1980s	10	1500–2000
WC4	Rural CDC built in 1980s	15	3000–4000
WC5	Urban clinic built after 2014	133	3000–4000
WC6	Rural CDC built after 2005	13	2500–3500

WC1–6, Western Cape clinic 1–6; CDC, community day care centre.

Within the selected clinics, we purposively sampled HCWs and facility staff who provided or supported TB-IPC services in different sectors. These staff included clinic managers (CMs) who supervised all healthcare activities in each facility; healthcare staff, including doctors, registered and enrolled nurses and HIV/AIDS, STI and TB (HAST) counsellors, facility administrators; and support staff, including clerks and cleaners.

### 2.2. Data Collection Methods

Twenty-three individual interviews and informal conversations were conducted with HCWs across the six clinics between May 2018 and June 2019. Another 25 HCWs participated in four FGDs. In WC3, two of the participants (professional nurse and cleaner) interviewed individually were also part of the group discussions. In WC5, two of the participants (Infectious Diseases Doctor and IPC Coordinator) were also part of the group discussions (see Table 2).

Individual interviews elicited information on the nature and provision of services provided to clients, current IPC plans and protocols regarding patient management, environmental controls, administrative, and governance practices. Participants were also asked about other staff roles and responsibilities, health worker protection, enablers and challenges to implementing IPC measures, and individual risk perception and risk management. Four focus group discussions (FGDs) of six to seven participants were held in three of the facilities (WC3, WC5 and WC6) to further explore enablers and challenges to implementing TB-IPC (See Table 2).

Each individual interview and FGD lasted between 45 min and 2 h while informal conversations lasted about 20–30 min. All interviews and group discussions were conducted in English, audio recorded and transcribed. When necessary, a multilingual research assistant conversant in isiXhosa and Afrikaans served as an interpreter for the first author, who led all interviews and group discussions.

**Table 2 ijerph-18-12133-t002:** Interviews, focus group discussions (FGDs), and role at the clinic.

Facility	Healthcare Worker	Role at the Clinic	Type of Data Collection
WC1	Professional nurse	Clinic manager	Interview
	Medical doctor	HIV/TB care	
	Professional nurse	TB nurse	
	Professional nurse	Chronic disease nurse	
WC2	Professional nurse	Clinic manager	Interview
	Professional nurse	HIV/TB Nurse	
	HIV/TB and STI (HAST) counsellor	HAST counsellor	
WC3	Professional nurse	Clinic manager	Interview
	Professional nurse	HIV/TB Nurse	
	Senior worker	Cleaner	
	HAST counsellor	HAST counsellor	
	Professional nurse	Clinic manager	FGD
	Enrolled nurse	Clerk	
	Support staff	Clerk	
	Professional nurse	TB nurse	
	Professional nurse	Childcare services	
	Senior worker	Cleaner	
WC4	Professional nurse	Clinic manager	Interview
	Professional nurse	HIV/TB nurse	
	Professional nurse	Mental health nurse	
	Professional nurse	Childcare services	
WC5	Professional nurse	Clinic manager	Interview
	Professional nurse	IPC coordinator	
	Medical doctor	Infectious diseases doctor	
	Administrator	Clerk	
	Professional nurse	TB nurse	
	Medical doctor	Infectious diseases doctor	FGD
	Pharmacist	Pharmacist	
	Professional nurse	IPC coordinator	
	Professional nurse	Operational manager–HAST programme	
	Administrator	Support services	
	Professional nurse	Outpatient manager	
	Dentist	Dentist in outpatient	
	Professional nurse	Nurse at outpatient	FGD
	Administrator	Receptionist	
	Professional nurse	TB and DR-TB nurse	
	Professional nurse	Chronic diseases nurse	
	Enrolled nurse	Infectious diseases nurse	
	Administrator	Clerk	
WC6	Professional nurse	Clinic manager	Interview
	HAST counsellor	HAST counsellor	
	Professional nurse	TB nurse	
	Professional nurse	Chronic diseases nurse	FGD
	Professional nurse	Outpatients nurse	
	Professional nurse	Chronic diseases nurse	
	Administrator	Outpatients nurse	
	Enrolled nurse	Integrated care nurse	
	Enrolled nurse	Integrated care nurse	

### 2.3. Data Analysis

Our analysis took a hybrid inductive-deductive approach, with Mannion and Davies’ overarching framework of organisational culture in health care settings providing a general deductive guide for interpretation of the data. A thematic networks approach was used to analyse the data [22,28]. We first used a set of broad codes to classify and organise the textual data. These codes included: community context; infrastructure; infection prevention and control; management; service delivery; and use of space. The broad coded segments helped us to then generate numerous themes; these were used to construct thematic networks linked to dimensions of organisational culture as put forward by Mannion and Davies [25], specifically visible manifestations (e.g., staffing, equipment, space, communication practices); shared ways of thinking (e.g., beliefs, values, and arguments put forward to sustain practice) and deeper assumptions implicit in dialogue and practice. We chose to focus on mask-wearing for this paper since this particular aspect of IPC most vividly illustrated both visible and explicit as well as less visible and implicit dimensions of organisational culture. As with all qualitative research, our objective in this study was not to develop generalizable quantitative claims about the prevalence of particular feelings, beliefs, behaviours, or experiences in a population, but rather to identify and explore some of the key patterns of thought, practice, relationship, and interaction that shape mask-wearing in this context.

### 2.4. Ethical Considerations

Ethical approval was secured from the University of Cape Town (Ref: 165/2018), the WC Provincial Government (Ref: WC_201806_001), the City of Cape Town (Ref: 23940) and the London School of Hygiene & Tropical Medicine (Ref: 14872). Before commencing with data collection, gatekeeper permissions from clinic managers were obtained. All participants signed an informed consent form and were informed that all information from the study would be kept confidential and anonymised.

## 3. Results

In the results below, we first look at the material and social dimensions of mask-wearing as an example of the visual manifestation of organisational culture. We then examine the normalisation and localization of TB risk as one expression of a shared way of thinking with regards to mask-wearing. Finally, we identify some of the deeper assumptions about responsibility and accountability, assumptions that are embedded in the power relations between HCWs and patients and that shape mask-wearing practices. Table 3 summarises the main themes and sub-themes we present in the results.

### 3.1. Material and Social Dimensions of Mask-Wearing: Visual Manifestations of Organisational Culture

For study participants, the experience of mask-wearing had important material and social dimensions. N95 respirators were felt to be much less comfortable to wear compared to surgical masks. They were tight, made participants hot, and made communication difficult. For some HCWs, the masks also had an unpleasant smell. Others also noted that the physical challenges of mask-wearing were greatly amplified in the context of poorly ventilated, crowded facilities. Although HCWs recognised that crowding could increase the risk of TB transmission, they, ironically, reported being less likely to wear masks in more crowded spaces. One nurse explained:

“I do not know if I’m using the right word, [it’s a] little bit stuffy, you see, so that’s a reason [for not wearing masks]… the space is not enough for these people with TB. Especially there in the morning, when they [patients] are too much there”.(Professional Nurse-WC3)

Several HCWs felt that masks created a barrier between them and the patients that they served. This was mostly spoken about in parts of the facility where patients were not perceived, by either staff or patients, to be infectious, places where they accessed treatment for non-communicable diseases, such as diabetes and high blood pressure:

“Not [wearing an N95 respirator] as much I should to be honest… because I think some patients do take offence if you cover your face while you’re busy with them because it’s, yeah, it’s a barrier. So, you can see they’re actually kind of offended if you put a mask on—or a mask when you are treating them”(Clinic Manager-WC6)

Others reflected on the differences of mask-wearing practice in South Africa in contrast with more routine mask-wearing in other countries:

“Look, I think simplistically-if you’re looking at overseas—if you’re looking at the Chinese or the Japanese—when they go outside, they’re all wearing a mask. It’s part of the norm. But here if I must go outside with a mask everyone will run. And it’s us that’s cultivated that—the only reason why you’re wearing a mask is because there’s something wrong with you. And that is the kind of stigma that we need to get away from”.(Doctor (FGD)-WC5)

Participants described the ways that masks could complicate their interactions with patients who might perceive their mask-wearing as a symbol of a negative view of patients. HCWs worried that patients would feel stigmatised if they wore masks while treating patients, and that masks created a social and communication “barrier” between them and their patients. In contrast, others spoke about the need for HCWs to normalise mask-wearing as a way to shift norms and practices and undercut the stigma that patients might experience when they wear masks (Doctor-WC5).

While many participants recognised the need for masks in crowded spaces and the opportunities that existed for reframing the social and moral significance of mask-wearing, the shared understanding in these clinics was that HCWs were not uniformly expected to overcome these challenges in their daily work.

### 3.2. Normalisation and Localisation of TB Risk: Shared Ways of Thinking

Although all HCWs interviewed were aware of the possibility that they could be infected with TB in the facilities where they worked, they held different views about the extent and nature of this risk. Some participants seemed resigned to the fact that they were likely to contract TB while at work. As one clinic manager put it:

“We all know we are at risk of contracting TB somewhere maybe in our careers”.(Clinic Manager-WC6)

Others worried more actively that the nature of their work and their interaction with patients who might have active TB could lead to their infection. Part of their concern stemmed from a fear of exposure to patients. HCWs expressed mistrust of patients who they saw as trying to conceal their TB or were themselves unaware that they had TB. Comments from one clinic manager highlighted this perceived risk:

“There is risk definitely because even those that we don’t know if they have TB already, they just walk in. They just come in saying they have flu with cough. We don’t know if that is TB…”.(Clinic Manager-WC1)

In addition to the uncertainty associated with not knowing patients’ TB status, participants also spoke about potential failures in the administrative control component of TB-IPC that could lead to nosocomial transmission of TB. Participants explained that the patient triage procedures that could decrease HCW risk of acquiring infection were not always closely followed. The policy in all the clinics was that patients who were diagnosed with TB/DR-TB were to be identified and fast-tracked at the clinic entrance. Not all of them were routinely detected at the entrance, however. In addition, participants noted that some patients might not be aware that their symptoms could be related to TB and may not report this on arrival.

While individual perceptions and explanations of TB risk varied, it was clear that the presence of TB risk was widely held to be a normal, even inevitable part of work in the clinic, rather than a risk worthy of urgent attention. TB risk was normalized as part of the unavoidable consequences of being an HCW. For some, TB should not necessarily be feared, either because they felt immune to infection after many years of exposure or because they had resigned themselves to the risk.

One way they seemed to manage the apparent contradiction—between knowing TB risk was present and that masks reduce that risk, on the one hand, and accommodating themselves to this risk and often not wearing masks, on the other—was by localising TB risks in certain places and people in the clinic. The HCWs we spoke with pinpointed particular areas within their facilities where they thought it necessary to wear a mask, for example, in spaces where TB patients were receiving treatment. The IPC nurse in one facility felt frustrated with what she saw as this ‘flawed’ perceptions among HCWs:

“The fact that you don’t wear mask in Emergency Unit, you don’t wear mask in the waiting room, you don’t wear mask in the outpatient department, means that you think TB is only in that room [TB room]…Even when you ask them to work there [TB room] somebody thinks ‘oh, I’m not going to get TB’. They do not realise TB you can get it whilst you are working here if you do not protect [yourself].(IPC Nurse-WC5)

Although participants knew that patients might not be aware that they had TB, several explained that their mask-wearing was based on their assessment of particular patients’ infectiousness, something that might not be grounded in reality. A facility doctor even described their own perception of personal protection as “very warped”:

“I do not wear a mask. I work here daily. I am in the room. I do not wear a mask. I will wear a mask, or I will be worried when I specifically see a patient. So, I do not deal a lot with the TB patients. But if I do see somebody that’s very sick then I will be a little bit strict with wearing the mask”.(Doctor-WC5)

These acts of locating risks in specific physical spaces within the facility or associating it with exposure to particular individuals emerged as an important shared, implicit knowledge in many facilities. It was not always clear whether HCWs really believed TB risk could so easily be located and managed, or they believed this inadequate approach was the only practical one available to them (or some mix of the two). Whatever the case, though, the localization of TB risks was a pervasive phenomenon in the clinics in this study. Even though all HCWs accepted the idea that mask-wearing could prevent TB infection, their perceptions of that risk as ‘normal’ or ‘inevitable’—rather than urgent and avoidable—and their belief that higher TB risk could consistently be located in TB-marked spaces, such as the TB waiting room, led many of them to wear masks inconsistently.

### 3.3. Deeper Assumptions Regarding Individual Responsibility versus Collective Good

One of the reasons HCWs were not pushed by managers or colleagues to wear masks more consistently or to challenge the stigma associated with masks was because of a deeply held idea among most staff that mask-wearing was, primarily, an individual responsibility. Some managers expressed the idea that it was an individual’s responsibility to prevent TB infection by wearing their N95 masks. When asked if HCWs knew about the risk of TB infection and how to minimise it, one facility manager responded:

“For TB, I think they, [the HCWs] perceive it as dangerous. But it’s the matter of—am I protecting myself enough? Am I doing what am I supposed to be doing? Because they know what they’re supposed to be doing…”.(IPC Manager-WC5)

Another responded:

“So yeah, we are at risk. But we must also make sure that we implement all the IPC precautions that are given to us because that is the other thing. If we didn’t implement our IPC or if I’m not looking at myself, I will be at risk. Do you understand what I mean?”(CM-WC4)

Some managers went on to explain that in the event an HCW did contract TB, they would immediately be held responsible, with specific reference to their use of PPE. One clinic manager said:

“And I tell them that remember, it is your own choice if you are going to use the mask or not, but remember if you get TB, the first thing they are going to ask is, ‘have you been using your mask?’”.(Clinic Manager-WC6)

Several clinic managers and doctors believed that they had an important role to play in shifting the culture that allowed for inconsistent mask-wearing. At the same time, however, they acknowledged that they also often failed to role model the kind of appropriate mask-wearing that they expected to normalise and support this practice:

“So, it’s very easy to tell a patient ‘no, you must wear a mask’ but if you’re not doing it yourself then who are we to educate them? So, it begins with us first. So, like I say if we are all wearing masks, the incidence of us contracting it as workers—not that I’m doing it myself to be very honest—but like I say it begins with us. So, we are looking at treatment whereas prevention by the person is less costly. So, we should be doing more to promote it. And I think that is where we are definitely lacking. If we are comparing ourselves to other countries”.(Doctor (NGT-FDG)-WC5)

Differing perceptions around the availability of masks also influenced whether mask-wearing was seen as an individual responsibility or more of a collective responsibility. HCW roles within the facility tended to shape views on the supply of masks. Some clinicians explained that due to a shortage in supply, they had to use a single N95 mask for a whole week instead of getting access to a new mask when they needed one.

“We have the masks obviously. Unfortunately, they are not changed every day—so we kind of change them once a week…Otherwise we run out. The supply is not enough for one person per mask per day”.(Doctor-WC3)

Clinic managers, on the other hand, were more often of the view that there were enough masks available, but that staff were simply not using them, or disposing of them too frequently.

“I would think people are just throwing those masks away. They [HCWs] are not reusing it [N95 mask]”(CM-WC3)

HCWs were thus not only blamed by some clinic managers for incorrect use of N95 masks, but also for wasting an important clinic resource. Finally, participants identified an additional challenge with respect to the often-vague protocols about how and when to wear or dispose of masks, as well as how to store them should they need to be reused. While there are national guidelines for HCW mask use, participants felt there was seldom enough guidance given to HCWs in different roles, for example, dentists who might need to observe slightly different protocols:

“There are no guidelines for us. So, I do not know… So, I am like blind. I just do what I think is the best for me to protect myself and my staff…I think of my discipline there are no protocols that’s specific to dentistry…”.(Dentist (FGD-WC5)

Participants also complained about the more general lack of coherence and oversight of TB-IPC guidelines within each facility. This was experienced as an indication of a lack of direction and urgency in relation to TB-IPC by management, again signalling that TB-IPC was something that individuals needed to make happen. Across the clinics we investigated, an underlying narrative of individual responsibility for mask-wearing was prominent, both explicitly, but also implicitly. The lack of role modelling, the under-supply of masks, and the vague and poorly enforced protocols around mask-wearing, for example, all communicated that wearing a mask was less a collectively held and enforced responsibility and more a matter of personal choice and agency.

## 4. Discussion

Our discussions with HCWs in WC facilities indicate that organisational culture has important impacts on mask-wearing and that there are important lessons to be learned about PPE use and IPC practices beyond our case study of TB in PHC clinics. HCW mask-wearing was shaped not only by individual knowledge and motivation but also by the embodied social dimensions of mask-wearing, the perceptions that TB risk was normal and localizable, and a shared underlying tendency to assume that mask-wearing, ultimately, was a matter of individual choice and responsibility.

The embodied and social dimensions of mask-wearing played a central role in HCW mask-wearing. The physical experience of wearing masks, especially N95 respirators, was associated with discomfort but also shaped social interactions among staff and between staff and patients. Crucial here is the perceived link between masks and stigmatised conditions, including TB, widely documented in South Africa and elsewhere [29,30]. Research in South Africa [14] found that newly diagnosed DR-TB patients felt stigmatised when they were asked by HCWs to wear N95 respirators. Masks can act as a symbol of the hidden assumptions about who is a danger that underlie a facility’s organisational culture. However, a study by Fix et al. [15] in a hospital setting found that social norms could also be a key driver driving mask-wearing. Organisational cultures are not fixed, and can be changed over time, to provide a more supportive environment of social norms for mask-wearing.

In the context of the high TB burden in South Africa, it is not surprising that the risk of TB infection has become ‘normalised’. Research on the development of TB-IPC policy in South Africa has described how the pervasiveness of TB triggers a sense of “fatalism” or “toughness” in HCWs and for those who have worked in the field for a long time, a sense of “invulnerability to infection” [31]. This sense of pervasiveness and inevitability can also make TB risk less visible and feel less urgent. In the Dominican Republic, a similar study spoke to how the “invisibility” of TB infection was a challenge in relation to enforcing TB-IPC [32]. Our findings suggested that risk was also imagined to be located within particular spaces and associated with particular individuals, a finding shared with another study on TB-IPC among nurses in South Africa [33]. The poor implementation of critical triage protocols at clinic entrances in this study, and others [4,5], also reflects this dangerous perception that TB risk can be known and predictably located in people and places already visibly marked as relating to ‘TB’.

Finally, we have explored the ways that mask-wearing is understood, both implicitly and explicitly, as the individual responsibility of HCWs. Some frontline HCWs and clinic managers articulated the idea that mask-wearing should be a collective responsibility that was enforced for the common good. However, this appeal was often undermined by other aspects of the organizational culture of clinics, including broader managerial and system failures, such as poor availability of masks, vague and poorly enforced protocols, and a general lack of role modelling and social norming around mask-wearing. These aspects of the daily practice and lived experience of working in a clinic could reinforce the message for frontline HCWs that they were largely on their own when it came to wearing masks.

Organisational culture—whether manifested in clinic protocols and the practices of HCWs, or more subtly, in the underlying beliefs and narratives that shape how HCWs understand the risks and responsibilities of their work—has an important, and under-researched, impact on HCW mask-wearing and other PPE and IPC practices. However, we also need more research on change in organizational cultures. As with culture more generally, organizational culture is not static, but changes over time, both through broader changes in the context as well as through the concerted effort of HCWs, managers, policymakers, and others. Some authors have usefully explored the relationships between aspects of organisational culture, such as leadership, and coordinated change efforts in health facilities and improved implementation of IPC [20,34]. Our participants, along with those in other studies [15,31] have likewise recognised that changes in mask use by opinion leaders within the clinic context could help to create a local cultural norm of mask-wearing.

Our findings suggest that mask-wearing in facilities would also be more consistent if facility managers more actively promoted engagement with TB-IPC guidelines and developed a sense of collective involvement and ownership of TB-IPC in facilities. If all HCWs wore masks all the time, patients might feel less stigmatised and in turn, be more likely to wear their own masks. Universal mask-wearing might also help to mediate the tensions between different kinds of HCWs, including facility management, by acting as powerful collective visual symbols of organisational culture that could ultimately highlight the health system’s support and care of HCWs by reducing risk of TB infection. This is, however, a challenging task as even in the Western Cape, with one of the best-managed and resourced provincial health systems in the country, mask-wearing is not consistent. This stands in contrast to many other areas of clinical practice where protocols or expectations are clear and non-negotiable.

There are some important limitations to this study. We did not begin this study with the aim of comprehensively assessing the impact of organisational culture on mask-wearing. As a result, we have not addressed other, likely important aspects of organisational culture, such as leadership roles, staff commitment to and satisfaction with their job, teamwork, health policy implementation and monitoring of staff performance. Another limitation of the study is our focus on the data from interviews and FGDs. Our interpretation would be enriched by including more of our ethnographic and observational data. This might deepen the analysis and also explain observed variations between clinics. Such an analysis, however, is beyond the scope of the initial interpretations offered in this paper.

## 5. Conclusions

This paper has explored the ways that organisational culture shapes mask-wearing for TB-IPC. Our data were collected prior to the heightened concerns around IPC in relation to COVID-19 that arose in early 2020. It is instructive to think about how our findings relate to these new circumstances. Despite the fact that the embodied and social challenges of mask-wearing are the same, mask-wearing in health facilities in relation to COVID-19 prevention has been very high in most settings, seen as both a public health intervention as well as a social practice [35]. This appears largely due to the fact that COVID-19 is perceived as an urgent and ‘unprecedented’ global health risk that is pervasive but can be effectively mitigated, and as a public health emergency in which mask-wearing is widely understood as a collectively responsibility and part of a common global good. These are not the perceptions that most HCWs bring to their work with TB. However, it may be the case that as the perceived risk and urgency of COVID goes down with vaccination, falling incidence, and better treatment, the same dynamics we have identified in this study will come into play. It is critical that we better understand how these dynamics can become so embedded in the organisational cultures of local health facilities, and how we might approach shifting embedded cultures and practices of infection prevention and control.

## Figures and Tables

**Table 3 ijerph-18-12133-t003:** Summary of themes and dimensions of organisational culture.

Themes	Dimension of Organisational Culture	Summary Explanation of Themes
Material and social dimensions of mask-wearing	Visual manifestation of organisational culture	N95 respirators were perceived to be less comfortable to wear. Masks were seen to create a barrier between HCWs and patients.
Normalisation and localization of TB risk	Shared ways of thinking	HCWs expressed that they felt more at risk of contracting TB while at work HCWs were concerned about patients who might have active TB that could lead to their infection. TB was seen to be located in particular areas within their facilities where it was necessary to wear a mask
Individual responsibility vs. collective good	Deeper shared assumptions	HCWs felt it was the individual’s responsibility to prevent TB infection by wearing their N95 masks. The failure to role-model the kind of appropriate mask-wearing practice, which may support this practice

## Data Availability

The data that support the findings of this study are openly available in Zivahub at zivahub.uct.ac.za (accessed on 1 November 2021).

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
