# Peer review of "Organisational Culture and Mask-Wearing Practices for Tuberculosis Infection Prevention and Control among Health Care Workers in Primary Care Facilities in the Western Cape, South Africa: A Qualitative Study"

_ijerph, 2021, doi:10.3390/ijerph182212133_

Round 1

Reviewer 1 Report

Dear Authors,

Thank you for the possibility to review your article. 

The article talks about the features of organisational culture in South Africa health care workers, with implications in the context of TB infection prevention and control.

The topic of the article is interesting but I think some implementations are needed. Here are some observations:

At line 55, the Authors should explain what they mean by medical masks, that they differentiate from surgical masks and N95 respirators.

In my opinion, the results should be presented after a thorough analysis instead of a descriptive analysis of the responses of individual health care workers. Moreover, it would be useful to have a schematization of the results according to the thematic areas encountered, accompanied by a statistical evaluation of the results; these could also be presented in the form of an explanatory table.

Kind regards

Author Response

Please see our responses to reviewers in the attached Word document. 

Reviewer 2 Report

This article concerns a pre-Covid study about organisational culture and mask-wearing practices for tuberculosis infection prevention and control in South Africa.  In my opinion, the paper needs a thorough revision. Some observations (not exhaustive):

  • Several typos in the text, especially repeated spaces, please review the entire document;
  • The abstract is not harmonious, some mention of the discussion is missing
  • Despite a well done interview process, structured according to specific guidelines deduced from the scientific literature, the sample size is too small to reach generalizable conclusions. It could be useful, after a review of the sample size, to try to make a synoptic view of the main proposals taken from the interviews, in a synthetic and tabular way. This table will then be useful to be recalled and reviewed step by step in the Discussion section;
  • The increase in the sample size will allow the homogenization of professional qualifications (each of which is not very representative, as there is often only one interview per qualification) and also of work experience;
  • PHC stands for? Please introduce here the acronym (Line 64);
  • Line 78-80 is not clear. Please better specify;
  • A small reduction of the introduction may be useful, the subdivision into paragraphs (paragraph 1.1) may be removed (Line 83);
  • Please, insert TB burdens data (Lines 130-131);
  • Lines 139-141: the sentence seems to belong to ‘Introduction’
  • Please, better specify how the enrolled facilities are (e.g., number of HCW employed, number of patients, services, etc. It might be useful to add a schematic table.
  • Limitations should be moved from Methods and put at the end of Discussion (Lines 193-202);
  • ”...The poor implementation of critical triage protocols at clinic entrances in this study...” where is specifically reported in the study? (Lines 415-416);
  • A useful data would be to understand if OHS training courses have been held, if this training has been performed focusing on the use of masks;
  • Lines 442-453: the statements contained in these sentences should be better supported by the results within the paper.
  • The messages deductively extrapolated from the study in support of the proposed objectives are not clear.

Author Response

(The authors gave the same response as above.)

Round 2

Reviewer 2 Report

In my opinion, the paper has improved according my suggestions. However the introduction still needs a revision because too long. It may be helpful to stress more in the text your thinking "... Whatever our final sample size, however, we do not believe our study is 'generalizable' in the conventional sense of the term in quantitative research. make precise quantitative estimates of how prevalent a particular feeling, experience or perspective may be, even within our sample .. " that you explain in your response.

Author Response

Please see attached Word Document.
